# Inhibition of RNA Helicase Activity Prevents Coxsackievirus B3-Induced Myocarditis in Human iPS Cardiomyocytes

**DOI:** 10.3390/ijms21093041

**Published:** 2020-04-25

**Authors:** Soo-Hyeon Yun, Ha-Hyeon Shin, Eun-Seon Ju, You-Jung Lee, Byung-Kwan Lim, Eun-Seok Jeon

**Affiliations:** 1Division of Cardiology, Samsung Medical Center, Sungkyunkwan University School of Medicine 50 Irwon dong, Gangnam-gu, Seoul 06351, Korea; shtrbyun@gmail.com (S.-H.Y.); ssony0115@hanmail.net (E.-S.J.); eelulee@daum.net (Y.-J.L.); 2Department of Biomedical Science, Jungwon University, 85 Munmu-ro, Goesan-gun 28024, Korea; hahyun925@naver.com

**Keywords:** coxsackievirus B3, helicase, myocarditis, enterovirus, cardiomyopathy, iPSC

## Abstract

Aims: Coxsackievirus B3 (CVB3) is known to be an important cause of myocarditis and dilated cardiomyopathy. Enterovirus-2C (E2C) is a viral RNA helicase. It inhibits host protein synthesis. Based on these facts, we hypothesize that the inhibition of 2C may suppress virus replication and prevent enterovirus-mediated cardiomyopathy. Methods and Results: We generated a chemically modified enterovirus-2C inhibitor (E2CI). From the in vitro assay, E2CI was showed strong antiviral effects. For in vivo testing, mice were treated with E2CI intraperitoneally injected daily for three consecutive days at a dose of 8 mg/kg per day, after CVB3 post-infection (p.i) (CVB3 + E2CI, *n* = 33). For the infected controls (CVB3 only, *n* = 35), mice were injected with PBS (phosphate buffered saline) in a DBA/2 strain to establish chronic myocarditis. The four-week survival rate of E2CI-treated mice was significantly higher than that of controls (92% vs. 71%; *p* < 0.05). Virus titers and myocardial damage were significantly reduced in the E2CI treated group. In addition, echocardiography indicated that E2CI administration dramatically maintained mouse heart function compared to control at day 28 p.i chronic stage (LVIDD, 3.1 ± 0.08 vs. 3.9 ± 0.09, *p* < 0.01; LVDS, 2.0 ± 0.07 vs. 2.5 ± 0.07, *p* < 0.001; FS, 34.8 ± 1.6% vs. 28.5 ± 1.5%; EF, 67. 9 ± 2.9% vs. 54.7 ± 4.7%, *p* < 0.05; CVB3 + E2CI, *n* = 6 vs. CVB3, *n* = 4). Moreover, E2CI is effectively worked in human iPS (induced pluripotent stem cell) derived cardiomyocytes. Conclusion: Enterovirus-2C inhibitor (E2CI) was significantly reduced viral replication, chronic myocardium damage, and CVB3-induced mortality in DBA/2 mice. These results suggested that E2CI is a novel therapeutic agent for the treatment of enterovirus-mediated diseases.

## 1. Introduction

Coxsackievirus B3 (CVB3) is identified as one of the most common etiological agents for heart diseases such as myocarditis and cardiomyopathy. It belongs to the picornavirus family and enterovirus genus. It has a positive single strand RNA genome. RNA directly translated to one polyprotein is soon infected. Therefore, CVB3 is replicated very quickly and kills infected host cells. Although most enterovirus illnesses are subclinical, acute myocardial inflammation can induce severe arrhythmias and sudden cardiac death. In addition, CVB3 infection may lead to the development of a chronic myocarditis and dilated cardiomyopathy (DCMP) [1,2,3,4]. CVB3 infection involves simple cold symptoms, such as fever and cough, and symptoms begin within seven days after infection. The gene structure and life cycle of CVB3 are well known [5,6,7,8,9]. However, effective vaccines and therapeutic drugs are not well-established, because RNA viruses easily change their capsid proteins and escape immune responses. Only limited treatment, such as IFN-gamma and immunoglobulin, is performed. To overcome this limitation, we studied various CVB3 inhibiting molecules. They showed antiviral effects in the experimental CVB3 acute murine myocarditis model. Specifically, IL-1 beta cytokine blocker, Coxsackie and Adenovirus receptor (CAR) trap inhibitor, 3C protease inhibitor (3CPI), and virus RNA polymerase inhibitor were studied. Each step of the virus life cycle can be applied for the targeting of antiviral agents [10,11,12,13]. 

RNA helicases play an essential role in a broad array of biological processes. During gene expression, RNA helicases catalyze ribonucleoprotein complex (RNP) rearrangements beginning at gene transcription and continuing during the consecutive steps in post-transcriptional gene expression: pre-mRNA splicing, mRNA export, translation, and turnover [14,15,16,17]. In the induction of the antiviral state, RNA helicases play a prominent role in the cellular response to viral infections. RNA helicases, both viral and human, recently emerged as novel targets for the treatment of viral infections [18]. Yedavalli showed that RNA helicase blocker significantly inhibited retrovirus replication [19].

In this study, we evaluated the antiviral activity of synthetic chemicals and chose E2CI (CVB3 RNA helicase 2C inhibitor) as the candidate to develop an antiviral agent against CVB3-induced viral myocarditis. E2CI significantly inhibited the replication of CVB3 and reduced heart damage and inflammations. It also preserved left-ventricular (LV) function and progression to DCMP in a murine viral myocarditis model. Moreover, the human IPS cardiomyocyte was completely protected from CVB3 infection by E2CI treatment. E2CI may be developed as a therapeutic drug for CVB3-induced myocarditis.

## 2. Results

### 2.1. Screen Effective Antiviral Agent

We generated chemical compounds based on viral protein structure [1]. The antiviral effects of various synthetic chemicals were observed using Renilla-luciferase expressing CVB3 (Renilla-CVB3) infection. HeLa cells were cultured in a 12-well plate, then chemicals were diluted and Renilla-CVB3 was added into the well plate. Following 24 h incubation, protein was extracted from the chemical treated HeLa cells in order to confirm Renilla-luciferase expression. Renilla-luciferase activity directly correlated with virus proliferation. SM1, SM2, and SM3 were suppressed Renilla-luciferase expression. In particular, SM1 (E2CI) showed a strong antiviral effect compared to Pleconaril (2A protease inhibitor), a positive control agent (Figure 1A). The chemical structure of E2CI was showed in Figure 1B.

### 2.2. E2CI Inhibit CVB3 Replication in HELA CELLS

The antiviral effects of E2CI were confirmed by GFP inserted CVB3 (GFP-H3) infection. HeLa cells were treated with either 10 ng/mL, 1 ng/mL, or 100 pg/mL E2CI with GFP-H3 for 24 h. The antiviral effects were observed by GFP expression under the fluorescent microscope. GFP expression was significantly decreased at 10 ng/mL of E2CI compared to without treatment (5.6 ± 0.5% vs. 42.3 ± 0.3% GFP positive cells, 10 vs. 0 ng/mL, *p* < 0.05) (Figure 2A). CVB3 replication was consistently increased at low dose of E2CI treatment. The CVB3 replication was directly observed by viral RNA amplification. CVB3 positive and negative strand RNA were significantly reduced through E2CI treatment in a dose-dependent manner (Figure 2B). There are no cytopathic effect observed with E2CI only treatment.

### 2.3. E2CI Decreases Mouse Mortality in a Murine Viral Myocarditis Model

E2CI in vivo effect was studied in a murine myocarditis model. Six-week-old male DBA/2 mice were intraperitoneally infected by 10^4^ pfu CVB3-H3 with or without E2CI treatment (8 mg/kg) from three days post-infection (p.i.) for three consecutive days. At days 5, 7, and 14 p.i., mice were sacrificed for organ virus titer and tissue inflammation measurement. Mice survival and heart function change were observed prior to the termination of the experiment at 28 days p.i. (Figure 3A). E2CI treatment improved mice survival rates compared to the untreated control group (CVB3 vs. CVB3 + E2CI, 70% vs. 95%, ***p* < 0.01) (Figure 3B). Heart and pancreas virus titer decreased in E2CI treated mice (Figure 3C). These data showed that E2CI inhibit virus replication in the subacute phase. Long-term mice survival rates were improved in the murine viral myocarditis model.

### 2.4. Decrease Cardiomyocyte Damage and Heart Inflammation

The heart histology was observed by H&E and Evans blue dye staining at 7 days post-infection. CVB3 infected mice hearts were damaged and inflammatory cell infiltrated into the dead cardiomyocyte areas. E2CI treatment significantly decreased cardiomyocyte death and inflammation compared to the untreated control group (CVB3 vs. CVB3 + E2CI, 23.67 ± 1.202 vs. 4.833 ± 1.327, *n* = 6) (Figure 4). E2CI also attenuated CVB3 replication in the cardiomyocytes and reduced heart damage.

### 2.5. E2CI Prevent LV Dysfunction and Progression of Cardiomyopathy in CVB3 Infection

Left-ventricular (LV) function was measured by echocardiography (Appendix A). The LV chamber diameter increased by CVB3 infection due to cardiomyocyte damage. E2CI treatment significantly reduced cardiomyocytes damage and preserved LV function. LV-systolic dimension (LVIDs) and diastolic dimension (LVIDd) were maintained better in the E2CI treated group compare to in the untreated control group (LVIDd, 3.1 ± 0.08 vs. 3.9 ± 0.09, *p* < 0.01; LVIDS, 2.0 ± 0.07 vs. 2.5 ± 0.07, *p* < 0.001; CVB3 + KR, *n* = 6 vs. CVB3, *n* = 4). In addition, fractional shortening (FS) and ejection fraction (EF) were dramatically preserved by E2CI treatment (FS, 34.8 ± 1.6% vs. 28.5 ± 1.5%, EF, 67.9 ± 2.9% vs. 54.7 ± 4.7%, *p* < 0.05; CVB3 + E2CI, *n* = 6 vs. CVB3, *n* = 4) (Figure 5). E2CI successfully inhibited CVB3 replication and prevented the LV dysfunction and development of dilated cardiomyopathy.

### 2.6. E2CI Inhibits CVB3 Replication in iPS Human Cardiomyocyte

The antiviral effect of E2CI was confirmed in the human iPS cardiomyocyte. GFP-H3 was infected with 10 ng/mL E2CI. The number of GFP expressing cells was significantly decreased in E2CI dose-dependent manner (1 ± 0.2% vs. 61.3 ± 0.4% GFP positive cells, 10 vs. 0 ng/mL, *p* < 0.05) (Figure 6A). In addition, virus gene amplification was strongly attenuated by E2CI treatment in both positive and negative strand RNA genome (Figure 6B). The cytotoxicity was not observed in human iPS cardiomyocytes by high concentration of E2CI treatment (Appendix A). These results strongly suggested that E2CI efficiently inhibits CVB3 replication in human cardiomyocytes. It should be used to develop an antiviral therapeutic drug for human myocarditis patient application.

## 3. Discussion

In this study, we established a new effective antiviral drug candidate from various synthetic chemicals for CVB3-induced myocarditis. E2CI was confirmed as a possible antiviral drug candidate. E2CI was injected intraperitoneally during the acute viremia phase significantly decreased virus replication and prevented progression from myocarditis to DCMP. In particular, E2CI also showed strong antiviral effects in the human IPS cardiomyocyte, respectively.

CVB3 belongs to the *Enterovirus* genus, the *Picornavirus* family, the same as poliovirus, and enterovirus71. It has a positive single strand RNA genome. Many vaccines have been observed against CVB3 and enterovirus71. Among them, no effective commercial vaccine developed yet, because RNA viruses very quickly change their surface antigens in order to escape from immune responses in our body. Therefore, we tried to search for effective antiviral molecules from synthetic chemical and natural compounds. It can be applied for the human patients after virus infection such as influenza therapeutic medicine *‘Tamiflu’*. CVB3 infection leads to direct cardiac injury. Although cellular immunity initiates acute myocarditis in Balb/C mice, myocyte-reactive antibodies are detected in the serum of infected animals during periods of inflammation and cardiac necrosis [1]. Before, we showed that early inflammatory cytokine receptor blocker and soluble coxsackievirus and adenovirus receptor (CAR) dimer have antiviral effects [10,11,13]. Pinkert et al., observed receptor protein antiviral effects using an adenovirus vector, but the use of a viral vector is limited for clinical applications [20].

Most antiviral agents have targets in the virus replication process related molecules. Virus receptor, RNA polymerase, virus protease, and virus packaging protein are very useful targets for restricting virus replication, propagation, and reinfection. Our previous study, we generated CVB3 replication inhibitor based on viral protein structure. Soluble 3C protease inhibitor (3CPI) inhibited virus replication, which was reflected in decreased virus titers in the heart and reduced myocardial damage. 3CPI treatment also prevented the progression from post-myocarditis remodeling to DCM [1]. In addition, 2A protease target inhibitor *‘Pleconaril’* was also developed as CVB3 therapeutic agents. However, this drug failed at in phase II clinical research. Helicases, both viral and human, recently emerged as novel targets for the treatment of viral infections [17]. Yedavalli showed that RNA helicase blocker significantly inhibited retrovirus replication [19]. E2CI is a novel RNA helicase inhibitor. It dramatically inhibited CVB3 replication and reduced myocarditis in the heart. However, we may need additional studies in order to define the mechanism of E2CI in direct regulating viral RNA helicase activity and the effect of the host gene replication in the later.

In this study, we found an RNA helicase inhibitor to be a CVB3 antiviral agent. E2CI was delivered intraperitoneally for three consecutive days after day 3 post-infection. It showed strong antiviral effects of markedly decreased virus titers, myocardial damage, and mouse mortality. In addition, E2CI treatment prevented LV dysfunction and the progression to DCMP. In addition, we observed the E2CI antiviral effect in human iPS cardiomyocyte. It is first time that CVB3 infection and antiviral effects in human cardiomyocyte were proven. These findings strongly suggest that E2CI may be developed as a new drug for cardiomyopathy associated with CVB3 infection.

## 4. Materials and Methods 

### 4.1. Virus and Cell Lines

CVB3 was derived from the infectious cDNA copy of the cardiotrophic H3 variant of CVB3. We also generated reporter gene Renilla-luciferase and green fluorescent protein (GFP)-inserted modified CVB3 (Renilla-CVB3 and GFP-CVB3) for the screening of effective chemicals. The virus titer was determined through plaque-forming assay and virus was amplified in a HeLa cell. The HeLa cell was cultured in 10% fetal bovine serum (FBS) contain Dulbecco’s Modified Eagle Medium (DMEM) culture media (Welgen, Seoul, Korea) [11].

### 4.2. Screening Antiviral Effect Chemical

We screened the antiviral effects of variant chemicals for coxsackievirus B3. The 2C inhibitor was identified based on a modified chemical structure of the 3C protease inhibitor. A HeLa cell was cultured in a 96-well plate for 16 h prior to virus infection. Then, HeLa cells were infected by Renilla-CVB3 with or without the modified chemical structure. After 24 h, 10 μL of Cell Counting Kit 8 (CCK8, Dojindo Lab, Tokyo, Japan) was added, and the cells were incubated for a further two hours. Cell survival was then measured with a multi-well plate reader at 450 nm. Non-infected HeLa cells were used as control and set arbitrarily to 100. Data is presented mean ± SEM from the three independent experiments.

### 4.3. Viral Myocarditis Murine Model

All procedures were reviewed and approved by the Institutional Animal Care and Use Committee of Samsung Biomedical Research Institute (SBRI, #20180227001). SBRI is accredited by the Association for Assessment and Accreditation of Laboratory Animal Care International (AAALAC International) and abides by the Institute of Laboratory Animal Resources (ILAR) guide. Five-week-old male DBA/2 mice were infected by intraperitoneal injection with 2 × 104 plaque-forming units (PFU) of CVB3 (Day 0). Mice were euthanized via cervical dislocation and sera and organs (heart, liver, and pancreas) were collected on days 7, 14, 21, and 28 [12]. Mice mortality was recorded, and virus titer was measured by PFU assay from the collected heart and pancreas. Cardiac damage was observed through histologic staining. E2CI antiviral effects were observed in the myocarditis murine model. Either E2CI 8 mg/kg in DMEM media or DMEM media alone (control) was injected from day 3 p.i. for three consecutive days (CVB3 + E2CI group, *n* = 35; CVB3 control group, *n* = 33). The mice were then euthanized and the sera and heart, liver, and pancreas were all collected, as described above. These mice (CVB3 control group, *n* = 4; CVB3 + E2CI group, *n* = 4 for each day) were excluded from the survival analysis.

### 4.4. Human iPS Cardiomyocytes Cell Culture

Highly purified human induced pluripotent stem cell-derived cardiomyocytes (iCell Cardiomyocytes^®^, Cellular Dynamics International, Inc., (CDI), Madison, WI, USA) were used in the experiments (Appendix A). HiPSC-CMs were seeded and maintained according to the manufacturer’s instructions, using iCell Cardiomyocytes Plating Medium and iCell Cardiomyocytes Maintenance Medium. Cryopreserved hiPSC-CMs were thawed and resuspended in iCell cardiomyocytes plating medium and plated on sterile 0.1% gelatin coated 24-well cell culture plate at a density of approximately 1.5 × 10^5^ cells/well. The cardiomyocytes were cultured in a cell culture incubator under culture conditions of 5% CO_2_ and 37 °C.

### 4.5. Viral Genome Amplification

The CVB3 RNA genome replication was confirmed by real-time PCR (polymerase chain reaction) for CVB3 capsid protein VP1 positive and negative strand RNA expression. Total RNA was isolated using TRIzol reagent (Invitrogen, Carlsbad, CA, USA). Reverse transcription reaction was performed for cDNA synthesis using a Maxime Reverse transcription (RT) kit (Intron Biotech, Inc., Seongnam, Korea) with 2 µL of the total RNA as the template. The real-time PCR was performed using RT cDNA as template, as well as VP1-sense primer (5′-GCGAAGAGTCTATTGAGCTA-3′) and VP1-antisense primer (5′-GTCAGCATGCGTGTACTTTA-3′) [1].

### 4.6. Histopathology and Organ Virus Titers

Heart and pancreas were collected for organ virus titer measurement at days 3, 7, 14, and 21 after CVB3 intraperitoneal infection. The basal parts of the hearts were homogenized in DMEM medium with 5% fetal bovine serum and the supernatant viral titers were determined by PFU assay. The apical parts of the hearts were fixed in 10% formalin, embedded in paraffin wax, sectioned at 5 µm, and finally stained with hematoxylin–eosin or picro Sirius-red and Von-Kossa staining. The sections were then graded for inflammation and the percentage area of myocardial fibrosis was evaluated using the NIH (National Institutes of Health) image quantification method [12].

### 4.7. Mouse Echocardiography

M-mode echocardiograms were performed using CVB3 infected mice. Adult mice were anesthetized with 1% isoflurane at day 28 p.i., and subjected to echocardiography. Echocardiography was performed under anesthesia with ketamine (100 mg/kg) and xylazine (5 mg/kg) mixture using a commercially available ultrasound system (Acuson Sequoi 512C system, Siemens, Mountain View, CA, USA) with a linear array transducer (14 MHz). The chest was shaved and the animal was positioned on a heating pad in a supine position. A single-channel electrocardiogram was obtained on the imaging system. Two-dimensional echocardiographic loops along with M-mode images of three consecutive beats were obtained [10].

### 4.8. Statistics

Data is presented as the mean ± SEM. The differences in measured parameters between the control and target groups were examined using the Mann–Whitney nonparametric *t*-test (Prism3.0 for Windows, GraphPad, La Jolla, CA, USA). Survival rates were analyzed using the Kaplan–Meier method, and *p* < 0.05 was considered to be statistically significant.

## 5. Conclusions

We found that E2CI showed strong antiviral effects of markedly decreased virus titers, myocardial damage, and mouse mortality. In addition, E2CI treatment prevented LV dysfunction and the progression to DCMP. In addition, it is the first time that CVB3 infection and antiviral effects in human iPS cardiomyocyte were proven. These findings demonstrate that E2CI may be developed as a new drug for cardiomyopathy associated with CVB3 infection.

## Figures and Tables

**Figure 1 ijms-21-03041-f001:**
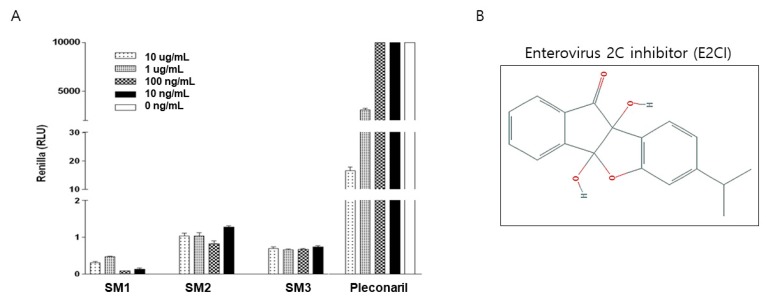
Screening antiviral effect chemical. (**A**) Anti-enterovirus activity of synthetic chemical was screened by virus replication inhibition assay in expression Renilla-luciferase following Renillar-CVB3 infection. 10 ug/mL, 1 ug/mL, 100 ng/mL, and 10 ng/mL concentration of synthetic chemical treatment reduced Renilla-luciferase expression (Relative luminometer units, RLU) compare to positive agent *Pleconaril*. SM1 (enterovirus-2C inhibitor (E2CI)) chose best candidate for antiviral chemical. (**B**) Chemical structure of E2CI.

**Figure 2 ijms-21-03041-f002:**
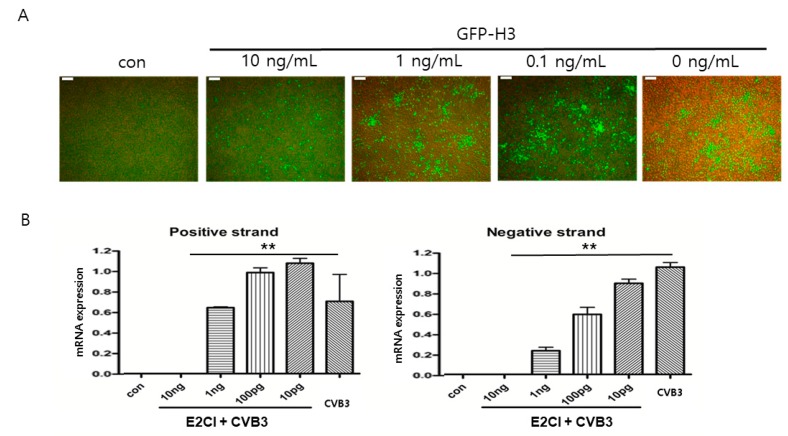
E2CI inhibit CVB3 replication in HeLa cells. (**A**) E2CI significantly inhibited CVB3 replication. Green fluorescent protein (GFP) was expressed during CVB3 replication with viral protein production. GFP expression was reduced by high dose (10 ng/mL) E2CI treatment (5.6 ± 0.5% vs. 42.3 ± 0.3% GFP positive cells, 10 vs. 0 ng/mL, *p* < 0.05). (**B**) CVB3 genome amplification was confirmed in CVB3 infected HeLa cells with E2CI treatment. CVB3 capsid protein VP1 gene positive and negative strand RNA were amplified by reverse transcription PCR. Both strand of VP1 RNA was significantly decreased by E2CI treatment. Data are presented as the mean plus or minus the standard error of the mean from three independent experiments. **, *p* < 0.01 (Scale bar, 100 μm).

**Figure 3 ijms-21-03041-f003:**
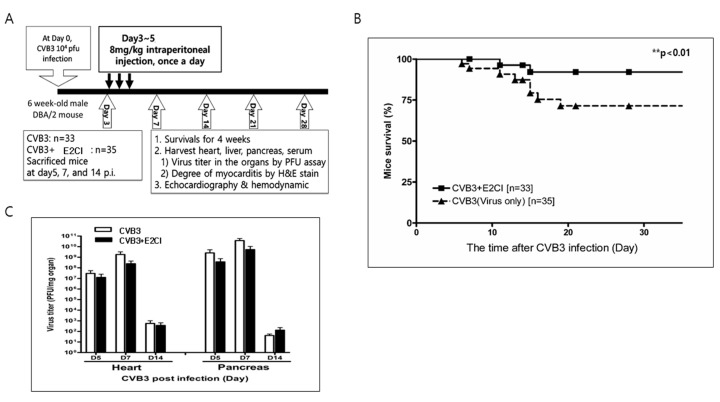
Decrease mortality and organ virus titer in murine myocarditis model. (**A**) In vivo experiment skim in murine viral myocarditis model. Tissue was corrected at Day 3, 7, and 14 p.i. for PFU assay and histological observation. (**B**) Mice survival was improved by E2CI treatment compare to untreated control group (CVB3 vs. CVB3 + E2CI, 70% vs. 95%, *p* < 0.01). (**C**) The live virus titer of the heart and pancreas were measured by PFU assay. E2CI decreased progeny virus production in the heart at day 7 p.i. Data are presented as the mean plus or minus the standard error of the mean from three independent experiments. **, *p* < 0.01.

**Figure 4 ijms-21-03041-f004:**
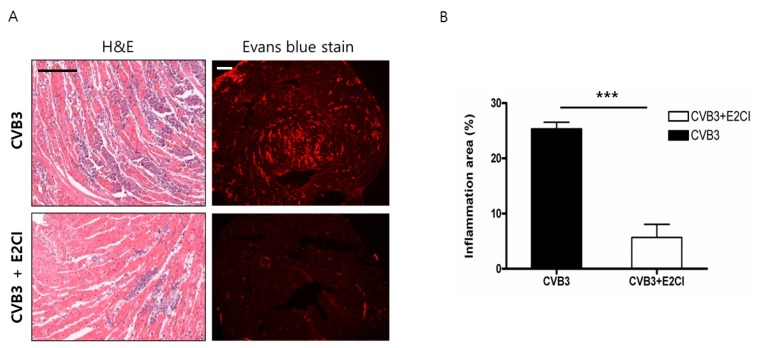
Histological finding and myocardium damage. (**A**) CVB3 infection induced heart damage. Inflammation and myocardium damage were observed by H&E and Evans blue stain at 7 days post-infection. Myocardium damage and inflammatory cell infiltration were significantly decreased by E2CI treatment. (**B**) Heart inflammation was quantified by imageJ software. E2CI treatment decreased inflammation area in the heart compare to untreated control group (CVB3 vs. CVB3 + E2CI, 23.67 ± 1.202% vs. 4.833 ± 1.327%, *n* = 6). Data are presented as the mean plus or minus the standard error of the mean from three independent experiments. ***, *p* < 0.001 (scale bar, 100 µm).

**Figure 5 ijms-21-03041-f005:**
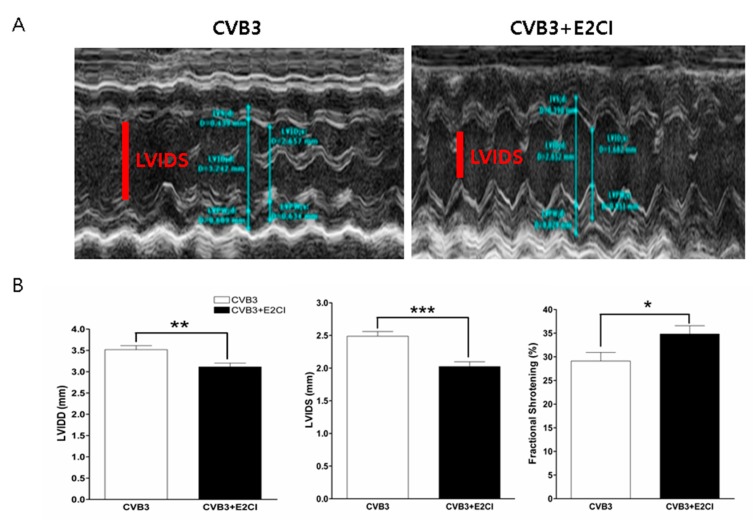
E2CI prevent LV dysfunction in viral myocarditis model. (**A**) M-mode echocardiography showed LV chamber dimension. E2CI treatment maintained heart function compare to untreated group. (**B**) Left Ventricular-systolic dimension (LVIDS, red bar) and diastolic dimension (LVIDD) were maintained in E2CI treated group compare to untreated control group (LVIDD, 3.1 ± 0.08 vs. 3.9 ± 0.09, *p* < 0.01; LVIDS, 2.0 ± 0.07 vs. 2.5 ± 0.07, *p* < 0.001; CVB3 + E2CI, *n* = 6 vs. CVB3, *n* = 4). Fractional shortening (FS) was dramatically preserved by E2CI treatment (FS, 34.8 ± 1.6% vs. 28.5 ± 1.5%, *p* < 0.05; CVB3 + E2CI, *n* = 6 vs. CVB3, *n* = 4). Data are presented from three independent experiments. ***, *p* < 0.001; **, *p* < 0.01, *, *p* < 0.05.

**Figure 6 ijms-21-03041-f006:**
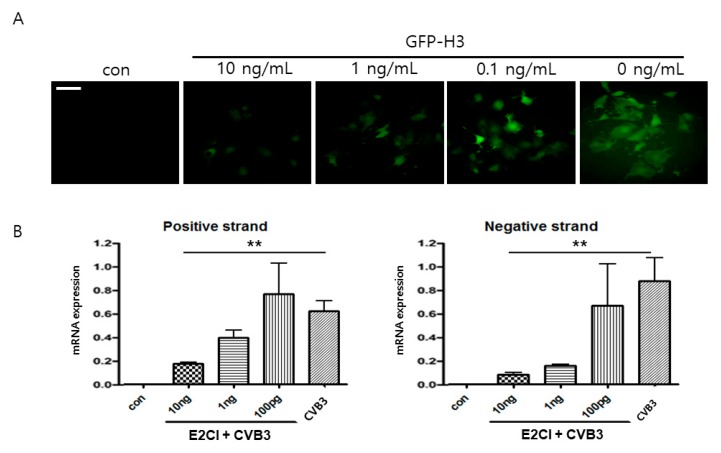
E2CI inhibit CVB3 replication in human cardiomyocyte. (**A**) E2CI significantly inhibited CVB3 replication with viral protein production. GFP expression was reduced by 10 ng/mL E2CI treatment in human iPS cardiomyocyte (1 ± 0.2% vs. 61.3 ± 0.4% GFP positive cells, 10 vs. 0 ng/mL, *p* < 0.05). (**B**) CVB3 positive and negative strand RNA were amplified by reverse transcription PCR. Both strand of VP1 capsid protein mRNA was significantly decreased by E2CI treatment. Data are presented as the mean plus or minus the standard error of the mean from three independent experiments. **, *p* < 0.01 (scale bar, 100 μm).

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
