# Peer review of "Inhibition of RNA Helicase Activity Prevents Coxsackievirus B3-Induced Myocarditis in Human iPS Cardiomyocytes"

_ijms, 2020, doi:10.3390/ijms21093041_

Round 1
Reviewer 1 Report
No further comments
Author Response
Thank you for your valuable comment.
Reviewer 2 Report
I still have concerns with the manuscript regarding the quality of the English used. Perhaps this will be cleared up during the publication process.
Author Response
Thank you for your valuable comment.
Reviewer 3 Report
The authors have been responsive to my critiques. They have added toxicity studies that help to demonstrate the antiviral effect of E2CI. In addition they have added better descriptions to the figure legends of the methods and included quantitation of key figures and significantly edited the manuscript. A minor point is that the toxicity studies added as supplemental data, do not include a figure legend and appear to be mislabeled (identical concentrations used in each panel). While they have added a sentence and reference pertaining to identification and testing of a protease inhibitor, I still feel that a more thorough description of the process that let to the identification of E2CI as a candidate inhibitor should be provided. Also, the authors should make clear that although this candidate was identified based on in silico analysis of 2C and it clearly inhibits the virus, there is no direct evidence that it actually inhibits this enzyme.
Author Response
The authors have been responsive to my critiques. They have added toxicity studies that help to demonstrate the antiviral effect of E2CI. In addition they have added better descriptions to the figure legends of the methods and included quantitation of key figures and significantly edited the manuscript. A minor point is that the toxicity studies added as supplemental data, do not include a figure legend and appear to be mislabeled (identical concentrations used in each panel).
--> We added figure legends in supplementary data.
While they have added a sentence and reference pertaining to the identification and testing of a protease inhibitor, I still feel that a more thorough description of the process that let to the identification of E2CI as a candidate inhibitor should be provided. Also, the authors should make clear that although this candidate was identified based on in silico analysis of 2C and it clearly inhibits the virus, there is no direct evidence that it actually inhibits this enzyme.
--> Thank you for your valuable comments. Previously, we developed the CVB3 3C protease inhibitor and reported it. We added as following: "2C inhibitor was identified base on a modified chemical structure of 3C protease".
But we have not generated CVB3 2C protease, so we could not show a direct regulating effect. We should define this in the later. We added the sentence in the discussion as following: “However, we may need additional studies in order to define the mechanism of E2CI in direct regulating viral RNA helicase activity and the effect of the host gene replication in the later.”